# Improvement of Wireless Localization Precision Using Chirp Signals

**DOI:** 10.3390/s25061844

**Published:** 2025-03-16

**Authors:** Ki-Tae Kim, Kwang-Yul Kim, Yoan Shin

**Affiliations:** 1School of Electronic Engineering, Soongsil University, Seoul 06978, Republic of Korea; kitae96@soongsil.ac.kr; 2Solid Wintech Inc., Yongin 16976, Republic of Korea; kimky@solidwt.co.kr

**Keywords:** chirp signals, oversampling-like effect, TDMA–tactical data link, chirp-based ranging algorithm, chirp-based localization algorithm, raised cosine interpolation, circular shift, TDoA, Fang’s algorithm

## Abstract

This study presents a novel localization framework that leverages the unique properties of chirp signals combined with a time division multiple access (TDMA)-based tactical data link to achieve high-precision positioning. Chirp signals, with their wide bandwidth and high temporal resolution, enable an oversampling-like effect, significantly enhancing distance estimation accuracy without the need for additional sampling rates. The proposed framework integrates chirp-based ranging and localization algorithms, incorporating raised cosine interpolation and circular shift techniques to improve temporal resolution and ensure precise peak detection. By utilizing the time differential of arrival (TDoA) and Fang’s algorithm, the system demonstrates robust performance, effectively mitigating challenges posed by multipath interference and jamming. The TDMA system provides synchronized time slots, allowing the seamless integration of communication and localization functionalities while ensuring stable and efficient operation. Experimental evaluations under various environmental conditions, including dense multipath and high-jamming scenarios, confirm the framework’s superiority over conventional localization methods in terms of accuracy, reliability, and resilience. These results highlight the framework’s potential applications in diverse fields, such as Internet of Things (IoT) networks, smart city infrastructure, and tactical communication systems, where high precision and robust localization are critical.

## 1. Introduction

Mobile wireless localization systems, such as Global Positioning Systems (GPSs), estimate the position of a mobile station (MS) using satellite signals, relying on precise timing information and satellite location data [1]. However, the accuracy of such systems significantly degrades in non-line-of-sight (NLoS) environments or areas with obstructions such as tall buildings or dense foliage. Differential GPS (DGPS) techniques can improve accuracy to within 10 m, while advanced algorithms like weighted least squares (WLS) are employed to mitigate errors and enhance localization precision [2]. In wireless sensor networks (WSNs), ranging-based methods such as time of arrival (ToA), time difference of arrival (TDoA), and received signal strength indicator (RSSI) provide high accuracy but often require precise synchronization or specialized hardware [3,4]. Alternatively, ranging-free methods, including centroid and distance vector (DV)-hop, offer cost-effective and energy-efficient solutions but suffer from reduced precision [5].

In order to improve the localization accuracy, a wider bandwidth and a high-rate sampling rate are needed. Thus, ultra-wideband (UWB), which uses a GHz-based sampling rate, is a representative scheme used to ensure the localization accuracy in centimeters, and an orthogonal frequency division multiplexing (OFDM)-based localization scheme to be used in 6G has also been studied [6,7]. However, since these studies concentrate on using the GHz-based bandwidth for the UWB and high-frequency band (i.e., THz) for 6G, the localization accuracy is limited on the distance reached outdoors, and the cost effectiveness is decreased by the GHz-based sampling rate. On the other hand, chirp signals have gained significant attention due to their ability to provide wide bandwidth and high temporal resolution by obtaining processing gain. These signals exhibit linear or nonlinear frequency modulation over time, enabling them to utilize a wide frequency band within the same time interval. This property increases the time–bandwidth product, allowing for high temporal resolution. Additionally, correlation processing enables the precise determination of time of arrival (ToA) and time difference of arrival (TDoA), making chirp signals advantageous for various localization applications. Chirp signals further exploit their wide frequency band to reduce signal intervals, enhance temporal resolution, and improve the signal-to-noise ratio (SNR), effectively separating multipath signals. These properties provide strong resistance to multipath interference and jamming, enabling reliable distance estimation and robust signal processing even in complex environments. Various studies have explored the use of chirp signals in wireless localization. TDMA has been integrated with UWB networks to improve mobile localization accuracy [8], while hybrid TDMA-FDMA acoustic localization systems have enhanced positioning through combined transmission schemes [9]. CSS-based localization has been applied to airborne acoustic positioning [10], and Bayesian filtering techniques have been used to improve accuracy and noise resistance [11]. These approaches demonstrate chirp-based localization’s effectiveness in mitigating multipath interference and enhancing signal resilience. To address these challenges, CSS has emerged as a promising technology for robust and long-range communication, featuring low probability of detection (LPD) and anti-jamming (AJ), making it suitable for IoT and LPWAN applications [12,13]. It is compatible with IEEE 802.15.4a [14] and has been widely implemented in frequency-hopping CSS (FH-CSS) and direct sequence spread spectrum (DSSS) systems [15,16]. CSS modulation techniques include direct modulation (DM-CSS) and binary orthogonal keying (BOK-CSS), balancing complexity and efficiency [17,18]. However, conventional CSS-based localization still faces challenges, with high sampling rate requirements and synchronization accuracy, leading to increased computational overhead and hardware complexity [19]. CSS, designed based on the intrinsic properties of chirp signals, maximizes processing gain, significantly enhancing signal detection and noise immunity. Utilizing a wide bandwidth and high temporal resolution, CSS mitigates jamming and multipath interference while reducing susceptibility to frequency-specific attacks and signal reflections. These features make CSS a reliable and effective solution for modern communication systems, especially in environments requiring high accuracy and robustness [20,21,22]. However, there has not yet been a study on improving the wireless localization accuracy of CSS signals, and CSS still has the disadvantage of low cost effectiveness because it considers a wider bandwidth and a higher sampling rate like conventional sampling-based ranging techniques.

Traditional CSS-based localization techniques using DM-CSS and BOK-CSS have inherent limitations in practical applications. DM-CSS, which modulates the phase of a single chirp signal, enables efficient transmission with phase shift keying (PSK) but increases receiver complexity due to the need for phase estimation and synchronization challenges. BOK-CSS, which maps up-chirp and down-chirp to binary symbols, simplifies receiver design by avoiding phase estimation but suffers from imperfect orthogonality, causing inter-symbol interference (ISI) and reduced localization accuracy. To address these challenges, this study proposes an enhanced chirp-based localization framework that can be applied to both DM-CSS and BOK-CSS to improve localization accuracy, while maintaining system efficiency. Unlike conventional CSS-based methods that rely on high sampling rates for better time resolution, the proposed framework employs raised cosine interpolation and circular shift techniques to achieve an oversampling-like effect without additional hardware complexity. These techniques enable precise TDoA estimation through correlation-based peak detection. Additionally, Zadoff–Chu sequence-based synchronization ensures reliable signal detection and mitigates synchronization errors in multipath and jamming environments. Furthermore, the proposed framework integrates seamlessly with a TDMA system, where synchronized time slots enable precise timing and stable performance without structural modifications. By applying these enhancements to DM-CSS and BOK-CSS, this approach improves localization accuracy and robustness, while preserving the benefits of each modulation scheme. A comparative analysis with conventional CSS-based methods highlights the advantages of the proposed framework, particularly in mitigating multipath interference and jamming effects.

The structure of this paper is as follows. In Section 2, we analyze existing research on TDMA-based tactical data link transceivers, focusing on previously developed studies, and propose a method to adapt them for wireless localization systems. Section 3 presents the design of a chirp-based ranging enhancement algorithm and a wireless localization algorithm. Section 4 evaluates the performance of the proposed wireless localization algorithm under various jamming environments and fading channels. Finally, Section 5 concludes the study.

## 2. Related Works

### 2.1. Analysis of TDMA-Based Tactical Data Link Transceivers

#### 2.1.1. Frame Structure Analysis

In TDMA-based tactical data links, the frame structure is designed to account for the operational characteristics of voice radios, including transmission and reception intervals. During transmission, a fixed delay known as the push-to-talk (PTT) attack time occurs when the PTT switch is activated, followed by RF signal transmission. Conversely, the PTT release time marks the cessation of data transmission when the PTT switch is deactivated, leaving this interval unused for communication [23,24].

In reception mode, automatic gain control (AGC) stabilizes the signal level to enable data demodulation. The AGC stabilization period, or AGC time, also temporarily halts data communication during this process. Prior frame structure designs incorporate the “PTT attack time”, “PTT release time”, and “AGC time”, allocating a PTT guard time of 210 ms for communication preparation, which includes both the “PTT release time” and “AGC time” [24,25].

Figure 1 illustrates the structure of a TDMA tactical data link frame. Each 300-millisecond time slot, containing 2400 symbols, is divided into four fields: the PTT guard field (1680 symbols or 210 ms), the synchronization field (288 symbols, including 256 synchronization symbols and 32 preamble symbols), the header field (32 symbols), and the data field (400 symbols). A TDMA tactical data link frame spans 12 s and supports up to 40 terminals, ensuring efficient resource allocation and communication management [25,26].

#### 2.1.2. Transmitter Analysis

The synchronization field in the frame structure consists of a total of 288 symbols, including 256 synchronization symbols and 32 preamble symbols. The synchronization symbols are generated as half-cycle waveforms at 4 kHz mapped to the bit “1” [19,20]. Previous studies adopted continuous phase frequency shift keying (CPFSK) modulation, mathematically defined as shown in (1) [20,27].(1)$$s_{k}\left(t\right)=a_{i}\sin(2\pi f_{i}\left(t-kT\right))$$
where $s_{k}\left(t\right)$ represents the transmitted CPFSK symbol, k represents the message bit index (k = {0, 1, 2,…,L−1}, where L is the total symbol length in each frame), i represents the message bit (i = {0,1}), $a_{i}$ is the sign determination factor, $f_{i}$ is the center frequency, and T is the symbol duration.

Based on (1), the transmitted signal corresponding to message bits can be expressed as in (2).(2)$$f_{i}=\left\{\begin{array}{c} 8000\;Hz,\;\;\;\;\;\;\;if\;d_{i}=1\\ \;\;\;4000\;Hz,\;\;\;\;\;\;\;if\;d_{i}=0\;\;\; \end{array}\right.$$
where $d_{i}$ is the message bit (“0” or “1”). For (1), the specific behavior of the data bits is further defined by (3):(3)$$a_{i}=\left\{\begin{array}{c} -a_{i-1},\;\;\;\;\;\;if\;d_{i}=0\\ \;\;\;a_{i-1},\;\;\;\;\;\;\;if\;d_{i}=1\;\;\; \end{array}\right.$$

The data bit “1” is mapped to a half-cycle waveform at 4 kHz, while the data bit “0” is mapped to a full-cycle waveform at 8 kHz. Additionally, depending on the current and previous bit states, a lookup table is employed to generate CPFSK waveforms, facilitating efficient mapping and symbol generation. This modulation technique ensures the stable transmission of tactical data in wireless environments, addressing challenges such as signal degradation or interference.

#### 2.1.3. Receiver Analysis

##### Preamble Detection

The zero crossing detection (ZCD) algorithm was applied to restore symbol timing. Figure 2 illustrates the accuracy of synchronization detection using the ZCD algorithm. The transmission signal magnitude, consisting of noise and the preamble, is represented in blue, while the output signal magnitude from the ZCD algorithm is shown in red. As the energy-per-bit to noise power spectral density ratio Eb/N0 increases from 5 dB to 15 dB, the accuracy of synchronization detection improves, enabling clear and precise synchronization detection [28,29]. By employing the ZCD algorithm, the reliability of synchronization detection is significantly enhanced, allowing more accurate signal detection even in challenging noisy environments.

##### Demodulation

Figure 3 illustrates the CPFSK demodulation process utilizing a non-coherent detection method.

In Figure 3, the received CPFSK signals $r\left(t\right)$ can be expressed as(4)$$r_{k}\left(t\right)=a_{i}(i)s_{k}(t)e^{j\theta }$$
where $a_{i}\left(i\right)$ is the received symbol coefficient representing the modulated signal amplitude and θ is the phase. This approach enables demodulation without relying on the phase information of the received signal, providing a simple yet efficient demodulation process. The non-coherent detection method ensures reliable demodulation performance across various environments, enhancing system robustness and versatility [20,27]. The values $z_{1}\left(T\right)$ and $z_{2}\left(T\right)$ are integrals used to extract the signal component at frequency $w_{1}$, while $z_{3}\left(T\right)$ and $z_{4}\left(T\right)$ serve a similar purpose for frequency $w_{2}$. The combined detection metric $z\left(T\right)$ is calculated as shown (5).(5)$$z\left(T\right)={z_{1}\left(T\right)}^{2}+{z_{2}\left(T\right)}^{2}-{z_{3}\left(T\right)}^{2}-{z_{4}\left(T\right)}^{2}$$(6)$$z_{1}\left(T\right)=\int_{0}^{t}r\left(t\right)\cdot \sqrt{\frac{2}{T}}\cdot cos\left(w_{1}t\right)dt$$(7)$$z_{2}\left(T\right)=\int_{0}^{t}r\left(t\right)\cdot \sqrt{\frac{2}{T}}\cdot sin\left(w_{1}t\right)dt$$(8)$$z_{3}\left(T\right)=\int_{0}^{t}r\left(t\right)\cdot \sqrt{\frac{2}{T}}\cdot cos\left(w_{2}t\right)dt$$(9)$$z_{4}\left(T\right)=\int_{0}^{t}r\left(t\right)\cdot \sqrt{\frac{2}{T}}\cdot sin\left(w_{2}t\right)dt$$

Finally, the received symbol ${\hat{S}}_{i}\left(t\right)$ represents the detected signal state based on $z\left(T\right)$, determining whether the state corresponds to $H_{1}$ or $H_{2}$.(10)$${\hat{S}}_{i}\left(t\right)=\left\{\begin{array}{c} H_{1},\;\;\;if\;z(T)\geq \;0\\ H_{2},\;\;\;if\;z(T)<0 \end{array}\right.$$

##### Frame Synchronization

The frame synchronization process involves performing a correlation operation with a 32-bit preamble. Frame synchronization is confirmed when the correlation exceeds a predefined threshold. Based on the detected frame synchronization signal, data recovery is performed. Figure 4 illustrates the results of CPFSK’s non-coherent detection and the preamble detection. The CPFSK demodulation results, obtained using the non-coherent detection, validate the simulation environment, as the performance matches that of uncoded binary frequency shift keying (BFSK) under conditions of accurate synchronization. Additionally, the preamble detection is accurately performed at the 257th symbol, immediately following the 256 synchronization symbols, thereby ensuring precise frame synchronization [20,25,27]. This analysis highlights the critical role of frame synchronization in the data recovery process. The successful detection of the preamble significantly contributes to improving the reliability of data recovery, ensuring robust performance even under challenging conditions.

### 2.2. Design of a Wireless Localization System Using Wireless Communication Frame Structures

The method for wireless localization within a TDMA-based tactical data link frame structure is illustrated in Figure 5. This process leverages the tactical data link frame configuration described in Figure 1 and proceeds as follows [23,24,25,26,30].

First, the master communication node synchronizes with GPS satellites. The master receives a 1 PPS (Pulse Per Second) signal from the GPS satellites, transmitted once every second, to align the system’s timing. This 1 PPS signal defines the overall frame time, which includes the PTT guard, sync, PHY header, and data fields.

Second, the slave communication nodes synchronize with the master node. The slaves detect the synchronization signal transmitted by the master and adjust their frame timing accordingly. This synchronization ensures that the slaves can transmit data within their allocated time slots (TS).

Third, the master detects the synchronization signals transmitted by the slaves. The master calculates the time difference between when each slave transmits the synchronization signal and when it is received by the master. This time difference is used to estimate the distance between the master and each slave.

Fourth, the ranging process involves the master measuring the time difference between the transmission and detection of the synchronization signals for each slave. For example, the distance between the master M and slave 1 (S#1) $S_{1}$ is calculated as [31,32](11)$$M-S_{1}=c\cdot {∆}_{1}/2.$$
where $c=3\cdot 10^{8}\;m/s$ represents the speed of light and ${∆}_{1}$ is the round-trip time of the signal. Similarly, distances between the master and other slaves are determined using the same approach.

Fifth, the tactical information transmission phase occurs. Each slave transmits data to the master during its designated transmission time based on the synchronization information it has measured. In this process, the differences between the synchronization signals and the transmission times are utilized to accurately estimate the precise positions of the slaves. This enables the master to calculate the location of all slaves simultaneously while ensuring accurate ranging and distance measurements.

Consequently, the above five steps demonstrate a method for achieving precise synchronization between the master and slaves using TDMA and accurately measuring the positions of the slaves. By maintaining synchronization through GPS signals and measuring distances to each slave, the master enables real-time wireless localization with high precision.

## 3. System Model

### 3.1. Chirp Spread Spectrum System

The CSS system uses chirp signals, which vary their frequency over time, to enable robust communication. CSS provides advantages such as resistance to multipath fading, efficient bandwidth utilization, and low energy consumption, making it widely applicable in radar, military, and commercial wireless systems [33,34].

CSS systems are primarily categorized into two modulation schemes: DM and BOK. DM transmits bits by altering the phase of a single chirp, offering efficient data transmission but requiring complex phase estimation. BOK, on the other hand, uses two distinct chirps (up-chirp and down-chirp) to represent bits, allowing non-coherent detection without phase estimation. While BOK slightly reduces performance compared to DM, it significantly simplifies receiver design [33,35].

### 3.2. Proposed Chirp-Based Wireless Localization System

Figure 6 illustrates the wireless localization system proposed in this study, which is designed based on chirp signals and consists of a transmitter and a receiver. This system is engineered to provide wide bandwidth and high temporal resolution, ensuring reliable location information even in jamming environments. Chirp signals exhibit a linear frequency variation over time, a characteristic that minimizes the impact of multipath fading and maximizes signal processing gain [34].

The proposed chirp-based wireless localization system is characterized by its high temporal resolution and robust resistance to jamming. It reduces receiver complexity while enabling precise location estimation. In particular, the system enhances temporal precision by employing synchronization signals based on the Zadoff–Chu sequences and utilizing raised cosine interpolation. Furthermore, the circular shift technique ensures flexibility in addressing all possible time offset scenarios. This design guarantees high reliability and accuracy in both military and commercial applications.

#### 3.2.1. Transmitter Design

The transmitter generates a preamble signal based on the Zadoff–Chu sequence, denoted as z(t). This preamble is incorporated into the transmitted signal to facilitate synchronization and time delay estimation at the receiver. The preamble signal, denoted as p(t), includes both up-chirp and down-chirp components to maximize the autocorrelation properties of the synchronization signal. In this system, chirp signals are transmitted according to the BOK method. Depending on the transmitted bit i, the chirp signal is either an up-chirp if i=1 or a down-chirp if i=0. Equation (12) represents the single linear chirp (SLC) signal $c_{i}\left(t\right)$ transmitted based on the bit i [33,35].(12)$$c_{i}\left(t\right)=Acos(2\pi f_{c}t-{\left(-1\right)}^{i}\pi \mu t^{2})$$
where $\left|t\right|\leq T_{c}/2$, $A=\sqrt{2E_{b}/T_{c}}$, $E_{b}$ represents the bit energy, $T_{c}=1/R_{b}$ is the chirp signal transmission duration, $R_{b}$ is the transmission rate, $f_{c}$ denotes the carrier frequency, $\mu =B/T_{c}$ is the chirp rate, and B represents the transmission bandwidth [28,30]. The wide bandwidth nature of the chirp signal provides high temporal resolution and ensures signal robustness in jamming environments. The transmission frame consists of a preamble and data, designed to follow a TDMA-based transmission structure [8,9,10,11].

#### 3.2.2. Jamming Model

In this system, both pulse jamming and tone jamming are considered. These jamming models assume that adversaries possess knowledge of the center frequency and bandwidth of friendly signals, leading to position estimation errors in friendly receivers due to adversarial GPS jamming.

The pulse jamming model exploits the property where short signals in the time domain appear widely distributed in the frequency domain. Pulse jamming applies strong, short-duration pulses to disrupt communication [36]. The mathematical model for this pulse jamming waveform $J_{p}\left(t\right)$ is expressed as follows [37,38].(13)$$J_{p}\left(t\right)=\sum_{n=-\infty }^{+\infty }g_{\tau }\left(t-nT\right)A_{J}cos\left(2\pi f_{J}t+\theta_{0}\right)$$
where $A_{J}$ represents the amplitude, τ is the width, T is the repeat time of the rectangular pulse jamming, $f_{J}$ is the carrier frequency, and $\theta_{0}$ is the initial phase of carrier.

In this study, it is assumed that all symbols are subjected to pulse jamming during the pulse duration. During correlation operations, the total received power is affected by the jammer state, which can be either active (“on”) or inactive (“off”). Specifically, when the jammer remains active during correlation, the state is referred to as jammer-on and denoted as $J_{on}$. Conversely, when the jammer is inactive, the state is referred to as jammer-off and denoted as $J_{off}$.

The single-sided power spectral density $N_{j}$ of the jamming signal is thus defined as follows [34].(14)$$N_{j}=\left\{\begin{array}{c} N_{j}/\rho \;\;\;\;\;\;\;\;with\;probability\;\rho \;(J_{on})\;\;\;\;\;\;\;\;\;\;\;\;\;\;\;\;\\ 0\;\;\;\;\;\;\;\;\;with\;probability\;\left(1-\rho \right)\;(J_{off})\;\;\;\;\;\;\;\;\; \end{array}\right.$$
where ρ represents the jammer’s duty cycle, which indicates the probability that a symbol is jammed. The duty cycle of the pulses per unit of time directly affects anti-jamming performance. Specifically, as the jammer’s duty cycle ρ decreases, multiple jamming signals interfere with a single symbol, thereby degrading the anti-jamming capability [36].

On the other hand, the tone jamming model generates a continuous single-tone signal at a specific frequency component. The mathematical model for this single-tone jamming waveform $J_{t}\left(t\right)$ is expressed as follows [38].(15)$$J_{t}\left(t\right)=\sqrt{2P_{J}}cos(2\pi f_{J}t+\theta_{J})$$
where $\sqrt{2P_{J}}$ represents the amplitude, $f_{J}$ is the carrier frequency, and $\theta_{J}$ is the phase, which is assumed to be uniformly distributed over [0,2π).

#### 3.2.3. Channel Model

The transmitted chirp signal propagates through the channel, undergoing the effects of multipath fading, noise, and jamming during the transmission process. The received signal r(t) can be expressed as follows:(16)$$y\left(t\right)=h\left(t\right)*r\left(t\right)+j\left(t\right)+n\left(t\right).$$
where $h\left(t\right)$ represents the channel impulse response, j(t) denotes the jamming signal, $n\left(t\right)$ denotes the noise, and the symbol “∗” indicates the convolution operation [39,40].

#### 3.2.4. Receiver Design

The receiver performs several signal processing steps to estimate the accurate signal arrival time and compute the position. First, the raised cosine filter is applied for time-domain interpolation to compensate for the mismatch between sampling intervals and propagation delay. This process generates virtual samples between the sampling intervals, significantly improving time resolution [41,42]. Second, a correlation operation is performed between the preamble signal and the received signal. During this step, a circular shift is applied to the synchronization signal to correct all possible time offsets, allowing the detection of the position with the maximum correlation value [43]. Finally, the signal arrival time is estimated based on the correlation results. Using the estimated arrival times, the TDoA between multiple anchor nodes is calculated [44], and the position is determined through Fang’s algorithm [45,46].

## 4. Chirp-Based Wireless Localization Algorithm

### 4.1. Design of the Chirp-Based Ranging Enhancement Algorithm

This section proposes a chirp-based ranging enhancement algorithm utilizing the TDMA-based tactical data link frame structure. The proposed method overcomes the limitations of traditional sampling-based ranging techniques, where discrepancies between sampling intervals and actual signal propagation delays result in significant localization errors [44]. Advanced signal processing techniques are applied to achieve the effect of oversampling [47].

Figure 7 illustrates the conceptual diagram of the proposed wireless localization system model. First, the master node transmits periodic synchronization signals (sync signals), synchronized via GPS or other timing references, to maintain time synchronization across the entire system. These transmitted synchronization signals provide a reference for the slave nodes to align their timing and serve as a preamble to enable the precise detection of signal arrival times.

Figure 8 illustrates a chirp-based ranging enhancement algorithm designed to improve distance estimation accuracy. The algorithm operates in offline and online phases.

#### 4.1.1. Offline Phase

The first step of the offline phase is storing time-interpolated chirp signals, which is the core concept of the proposed scheme. In the first step, the chirp signal with raised cosine interpolation is applied to enhance the time resolution by increasing the number of samples according to the oversampling rate [41,42]. Then, the one-sample circular shift technique is used to generate high-resolution time-delay states by cyclically shifting the oversampled signal at a certain time [43]. However, the system has to use the ADC sampling rate for ranging, and the sampling step samples the oversampled signal by the ADC sampling rate and stores the buffers by the number of the oversamples. These steps result in oversampling-like effects, even though the system is at the same sampling rate as the ADC’s and ensures finely tuned signals, enabling precise peak detection in the later stages. Figure 9 illustrates an example for sampling in the offline phase, as explained above.

#### 4.1.2. Online Phase

The online phase involves two steps: coarse ranging and fine ranging. Coarse ranging provides an initial estimation of signal propagation delay using correlation operations at the analog-to-digital converter (ADC) sampling rate. While effective, this step may introduce errors due to sampling intervals or interference. Fine ranging refines the estimation by analyzing interpolated signals from the offline phase, starting slightly earlier than the coarse-detected peak time. This approach compensates for timing errors and accurately identifies the signal arrival time, significantly improving distance estimation precision [48,49].

The core of the algorithm is peak detection through correlation operations using the Zadoff–Chu sequence as a synchronization reference. Correlation identifies the maximum similarity between the received signal and the predefined preamble, ensuring precise signal arrival time estimation [50].

The algorithm leverages the CSS technique, which spreads the signal over a wide frequency band to enhance the SNR, resist interference, and maintain temporal precision. The processing gain of CSS ensures robust performance by protecting against noise and jamming, mitigating multipath effects, and improving temporal resolution. These capabilities make the proposed algorithm highly effective for precise wireless localization, even in challenging environments [20].

The Zadoff–Chu sequence, with its unique correlation properties, enables precise wireless localization by ensuring accurate peak detection during correlation operations. Its constant amplitude and low side-lobe autocorrelation minimize errors, even in noisy or interference-prone environments. The near-zero cross-correlation with other signals preserves the sequence’s integrity, producing sharp and distinct peaks for precise measurement of ToA or TDoA. This makes the Zadoff–Chu sequence highly effective for reducing localization errors and achieving reliable position estimation, even in challenging multipath fading conditions [51,52].

### 4.2. Design of the Chirp-Based Localization Enhancement Algorithm

This section considers the TDMA-based wireless localization algorithm presented in Figure 8 and applies the TDoA-based Fang’s algorithm [45,46] to perform wireless localization using the ranging results. Fang’s algorithm provides a direct solution to localization by matching the number of TDoA-based distance measurement equations with the number of unknown parameters, specifically the coordinates of the mobile station (MS). Unlike iterative approaches such as the Taylor series expansion [53], which require repeated calculations to find approximate solutions, Fang’s algorithm is computationally efficient and solves the localization problem with a straightforward process.

However, Fang’s algorithm is limited to using only three reference nodes for positioning, which restricts its ability to further improve accuracy with additional nodes. By leveraging the TDoA values of these three reference nodes, the algorithm determines the location of the target mobile station efficiently. The calculation process is as follows.

Assume that the location of the target node is (x, y) and the position of the reference node ${BS}_{i}$ is ($x_{i},\;y_{i}$)$\;\left(i=1,2,3\right)$; $D_{i}$ is the difference between the target and the reference node, which can be represented as(17)$$D_{i}=\sqrt{{\left(x_{i}-x\right)}^{2}+{\left(y_{i}-x\right)}^{2}}$$

According to the TDoA described in (18)(18)$$D_{i}^{2}={\left(D_{i,1}+D_{1}\right)}^{2}$$(19)$$D_{i,1}=D_{i,1}-D_{1}=\sqrt[{}]{{\left(x_{i}-x\right)}^{2}+{\left(y_{i}-x\right)}^{2}}-\sqrt[{}]{{\left(x_{1}-x\right)}^{2}+{\left(y_{1}-x\right)}^{2}}$$
where (18) can be expanded as(20)$$D_{i,1}^{2}+2D_{i,1}D_{1}+D_{1}^{2}=x_{i}^{2}+y_{i}^{2}+x^{2}+y^{2}-2x_{i}x-2y_{i}y$$

According to (17) and (20), we have(21)$$D_{i,1}^{2}+2D_{i,1}D_{1}+D_{1}^{2}=K_{i}-K_{1}-2x_{i,1}x-2y_{i,1}y,$$
where $K_{i}=x_{i}^{2}+y_{i}^{2}$, $x_{i,1}=x_{i}-x_{1}$, and $y_{i,1}=y_{i}-y_{1}$. Actually, in order to simplify the calculation, the coordinates of BS1 are set to (0, 0) and BS2 is set $\left(x_{2},\;0\right)$; then, (21) can be simplified as(22)$$D_{i,1}^{2}+2D_{i,1}D_{1}=K_{i}-2x_{i}x-2y_{i}y$$

From (22), we have for BS1 and BS2,(23)$$D_{2,1}^{2}+2D_{2,1}D_{1}=K_{2}-2x_{2}x-2y_{2}y$$

Similarly, for BS1 and BS2,(24)$$D_{3,1}^{2}+2D_{3,1}D_{1}=K_{3}-2x_{3}x-2y_{3}y$$

Bu substituting (23) into (24), we can obtain(25)y=kx+b
where $k=(x_{2}D_{3,1}/D_{2,1}-x_{3})/y_{3}$ and $b=\left[K_{3}-D_{3,1}^{2}+D_{3,1}D_{2,1}\left(1-K_{2}/D_{2,1}^{2}\right)\right]/2y_{3}$(26)$$dx^{2}+ex+f=0$$
where $d=1+k^{2}-{\left(x_{2}/D_{2,1}\right)}^{2}$, $e=2kb+x_{2}(\frac{K_{2}}{D_{2,1}^{2}}-1)$, and $f=b^{2}-\frac{{\left(K_{2}-D_{2,1}^{2}\right)}^{2}}{4D_{2,1}^{2}}.$

Generally, there are two solutions to $x=\frac{-e\pm \sqrt{e^{2}-4df}}{2d}$ when solving (26); however, the invalid solution can be excluded according to prior information, and then the location of the target (x, y) can be resolved through (25).

## 5. Experimental Results and Analysis

### 5.1. Ranging Performance

This section evaluates the performance of the proposed chirp-based ranging enhancement scheme under various conditions, including the additive white Gaussian noise (AWGN) channel and jamming environments (pulse and tone jamming). The performance is compared against the conventional sampling-based ranging scheme to highlight the advantages of the proposed approach. In order to evaluate the performance, the root mean squared error (RMSE) is calculated as follows.(27)$$RMSE(\hat{x})=\sqrt{\frac{1}{M_{c}}\sum_{m=1}^{M_{c}}{\left||x^{\left(m\right)}-{\hat{x}}^{\left(m\right)}|\right|}^{2}},$$
where $M_{c}=10,000$ is the number of Monte Carlo runs.

Figure 10 and Figure 11 show the comparison results, with Figure 10 depicting performance in ideal no-noise (left) and AWGN (right) conditions and Figure 11 illustrating performance under pulse jamming (left) and tone jamming (right) scenarios. Table 1 summarizes the key parameters used for performance evaluation.

Under an ideal no-noise case, the conventional sampling-based ranging scheme demonstrates positioning errors up to 9 m, caused by mismatches between the sampling interval and propagation delay [54,55]. As mentioned earlier, in order to improve the ranging accuracy, UWB, which is a representative scheme of sampling-based ranging, is considered [6,7]. However, since UWB needs a GHz-based sampling rate, the localization system must be equipped with more expensive ADC chipsets to ensure the localization accuracy. The proposed scheme addresses this issue using raised cosine filtering for time-domain interpolation, significantly enhancing time resolution and mitigating these mismatches. Please note that the proposed scheme obtains the localization accuracy the same way as a high-speed ADC sampling rate even though it is in a low-speed sampling rate. This is the reason that the chirp signal changes the instantaneous frequency according to time for spreading the signal bandwidth, and the ranging resolution can be improved by not only a time–frequency product, known as processing gain, but also time-interpolated chirp signals, which can extract a sample more sensitively. In order to improve the ranging resolution for the proposed scheme more accurately, the time–frequency product should be more increased, and the time-interpolated chirp signals should be more oversampled. Since the proposed scheme spreads the spectrum from 8 kHz to 1 MHz and uses time-interpolated chirp signals, the processing gain is obtained as 125 times better, and the ranging resolution is guaranteed at an RMSE level of below 1 m even though the chirp bandwidth is 1 MHz. As shown in Figure 10 (left), the proposed scheme achieves consistently lower errors than the conventional approach.

In the AWGN channel, both schemes utilize Zadoff–Chu sequence preambles of lengths 128 and 256. The proposed scheme leverages the CSS processing gain and Zadoff–Chu correlation properties to achieve a 5 dB improvement in ranging accuracy over the conventional scheme with a sequence length of 256. Additionally, at higher $E_{b}/N_{0}$ levels, the proposed scheme delivers an extra 6.5 dB improvement, showcasing robustness in noisy environments, as seen in Figure 10 (right).

Under pulse jamming conditions, the proposed scheme, with a Zadoff–Chu sequence of length 256, achieves a 4 dB improvement in ranging accuracy compared to the conventional scheme, along with an additional 5 dB improvement at higher $E_{b}/N_{0}$ levels. This enhanced performance is attributed to CSS’s wideband spreading, which reduces susceptibility to short-duration jamming and enables reliable data recovery, as shown in Figure 11 (left).

In tone jamming scenarios, the proposed scheme demonstrates a 5.5 dB improvement in accuracy with a Zadoff–Chu sequence of 256 length and an additional 6 dB improvement at higher $E_{b}/N_{0}$ levels. The wideband spreading characteristics of CSS mitigate interference concentrated at specific frequencies, ensuring only small portions of the signal are affected. This capability, combined with CSS’s processing gain, allows the proposed scheme to consistently outperform the conventional approach under tone jamming conditions, as illustrated in Figure 11 (right).

### 5.2. Wireless Localization Performance

This section evaluates the performance of the proposed chirp-based wireless localization scheme under various channel conditions, including jamming (AWGN, pulse jamming, and tone jamming) and fading (Rayleigh and Rician) environments, as well as combined jamming and fading scenarios. The results compare the proposed scheme with the conventional sampling-based localization scheme, highlighting its advantages in handling challenging interference and fading effects. Figure 12 and Figure 13 present the localization error performance across these scenarios, with Figure 12 illustrating results under jamming conditions (left) and fading environments (right) and Figure 13 showing results under combined pulse jamming and fading conditions (left) and combined tone jamming and fading conditions (right). Table 2 summarizes the key parameters used for evaluation.

Under jamming conditions, the conventional scheme exhibits substantial localization errors exceeding 50 m at low $E_{b}/N_{0}$ levels due to mismatches between the increased sampling interval and the signal propagation delay. Even at higher $E_{b}/N_{0}$ levels, errors remain above 50 m. In contrast, the proposed scheme effectively mitigates these challenges. In pulse jamming environments, localization errors are reduced to within 20 m at low $E_{b}/N_{0}$ levels and further decrease to 10 m or less at higher levels. Similarly, in tone jamming environments, errors are reduced to approximately 15 m at low $E_{b}/N_{0}$ levels and to 5 m or less at higher levels. These improvements are attributed to the wideband spreading properties of CSS, which mitigate the effects of short-duration and frequency-specific interference and the strong correlation characteristics of the Zadoff–Chu sequence, which enable precise signal arrival time estimation, as shown in Figure 12 (left).

In fading environments, the conventional scheme demonstrates poor performance, with localization errors ranging from 50 to 60 m under Rayleigh and Rician fading. At low $E_{b}/N_{0}$ levels, errors exceed 60 m, with minimal improvement even at higher levels due to severe time mismatches and signal strength fluctuations. The proposed scheme, however, achieves robust performance. Localization errors are reduced to approximately 20 m under Rayleigh fading and 15 m under Rician fading at low $E_{b}/N_{0}$ levels. At higher levels, errors decrease to within 5 m for both fading environments. This superior performance is achieved through the strong correlation properties of the Zadoff–Chu sequence and the resilience of CSS to multipath fading and interference. The presence of a line-of-sight path in Rician fading further enhances localization accuracy, as illustrated in Figure 12 (right).

Under combined jamming and fading conditions, the conventional scheme suffers from significant localization errors. In pulse jamming combined with fading, errors range from 50 to 75 m at low $E_{b}/N_{0}$ levels, with negligible improvement as $E_{b}/N_{0}$ increases. Similarly, in tone jamming combined with fading, errors range from 55 to 65 m. The proposed scheme, however, demonstrates robust performance even in these challenging conditions. Under pulse jamming with Rayleigh fading, errors are reduced to approximately 25 m at low $E_{b}/N_{0}$ levels and further decrease to within 15 m at higher levels. With Rician fading, errors improve from 20 m to within 10 m, as shown in Figure 13 (left). Similarly, under tone jamming with Rayleigh fading, errors decrease from 20 m to within 5 m, and with Rician fading, they improve from 15 m to within 5 m at higher $E_{b}/N_{0}$ levels, as illustrated in Figure 13 (right).

The proposed scheme’s superior performance is attributed to its integration of CSS and Zadoff–Chu sequences. CSS mitigates jamming effects by spreading the signal across a wide frequency range, ensuring that pulse jamming impacts only limited portions of the time–frequency domain, while tone jamming, localized to narrow frequency bands, affects only small portions of the signal. The strong correlation properties of the Zadoff–Chu sequence further enhance signal arrival time estimation accuracy under compounded interference.

Overall, the proposed chirp-based localization scheme consistently achieves lower localization errors across all scenarios. Its robustness and high accuracy make it a significantly reliable and effective solution for operational environments that require precise and resilient localization under challenging conditions.

## 6. Conclusions

This study proposed a localization framework leveraging the robust characteristics of chirp signals and the processing gain achieved through CSS, offering a novel approach that delivers exceptional precision and reliability even in complex and challenging environments. The broadband nature of chirp signals and their oversampling-like effect enable accurate TDoA measurements, significantly enhancing the accuracy of distance and position estimation. Furthermore, the proposed framework demonstrates strong resilience to multipath interference and jamming, ensuring robust localization performance under adverse channel conditions.

The integration of the chirp-based localization framework with a TDMA system enhances synchronization and enables the seamless operation of both localization and communication functionalities within a single framework. The TDMA structure facilitates precise timing synchronization, optimizing the temporal resolution capabilities of chirp signals and ensuring stable data transmission. This integration not only reduces system complexity but also maximizes both communication and localization accuracy, presenting significant potential for applications in IoT networks, smart cities, autonomous vehicle systems, and tactical networks.

The proposed framework also incorporates advanced techniques such as raised cosine interpolation, circular shift operations, and Zadoff–Chu sequences to improve temporal resolution and enable precise peak detection in correlation operations. Experimental evaluations conducted under various channel conditions, including jamming and multipath interference, demonstrate the algorithm’s superiority in maintaining low localization errors, outperforming conventional methods in terms of both accuracy and robustness.

This research has theoretically and experimentally validated the proposed framework’s superior performance compared to existing approaches, establishing a foundation for next-generation localization technologies.

Future work will focus on validating the proposed methods in hardware-based environments to confirm their real-world performance and expand the system’s applicability to broader network conditions and operational challenges. Further optimization of the framework will aim to ensure consistent high performance in diverse and demanding scenarios.

This study highlights the potential of chirp-based localization technologies as a transformative solution for precise localization while also demonstrating the proposed framework’s significant contributions to a wide range of applications requiring high accuracy and resilience.

## Figures and Tables

**Figure 1 sensors-25-01844-f001:**
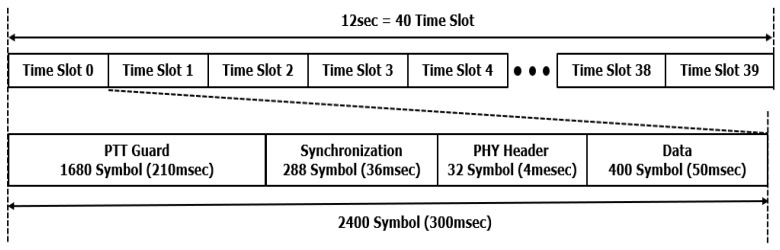
Tactical data link frame structure.

**Figure 2 sensors-25-01844-f002:**
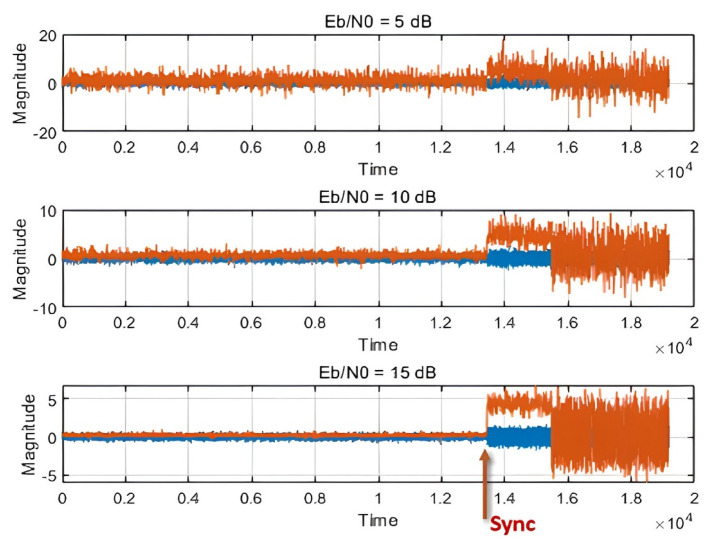
Synchronization detection using the ZCD algorithm.

**Figure 3 sensors-25-01844-f003:**
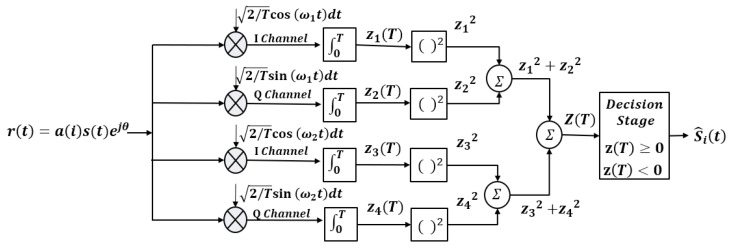
CPFSK demodulation process.

**Figure 4 sensors-25-01844-f004:**
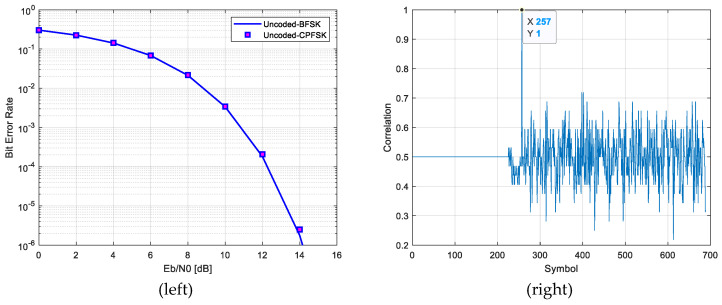
CPFSK’s non-coherent detection results (**left**) and preamble detection results (**right**).

**Figure 5 sensors-25-01844-f005:**
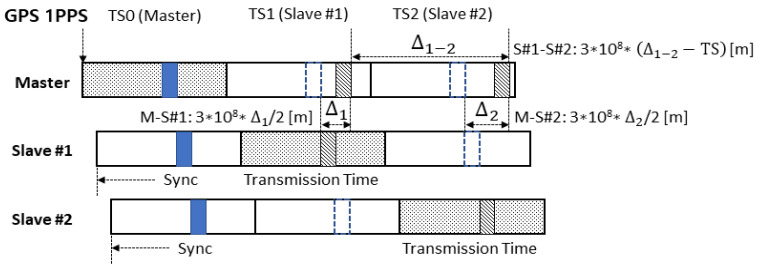
Method for wireless localization using TDMA.

**Figure 6 sensors-25-01844-f006:**
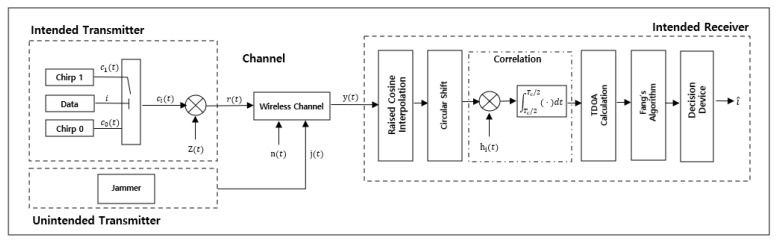
Proposed chirp-based wireless localization system.

**Figure 7 sensors-25-01844-f007:**
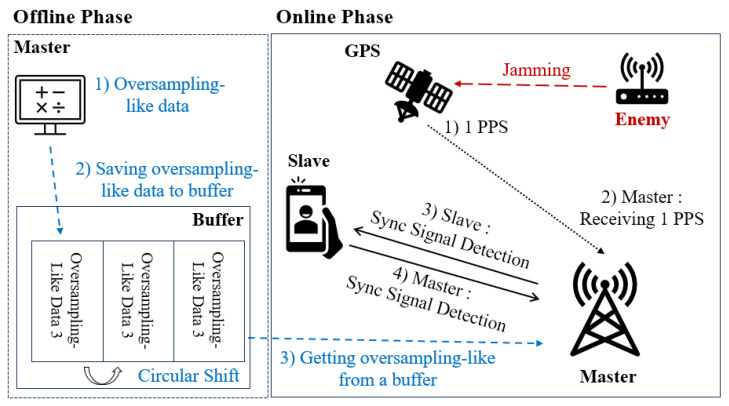
Conceptual diagram of the proposed wireless localization system model.

**Figure 8 sensors-25-01844-f008:**
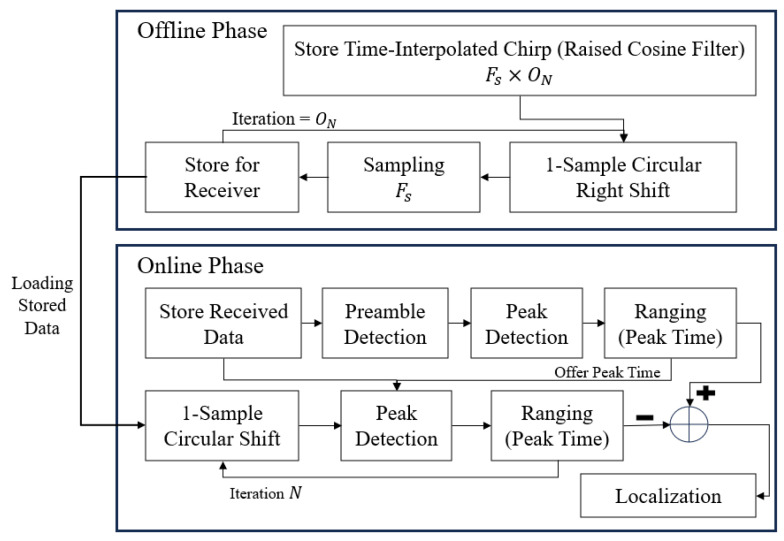
Proposed chirp-based ranging algorithm structure.

**Figure 9 sensors-25-01844-f009:**
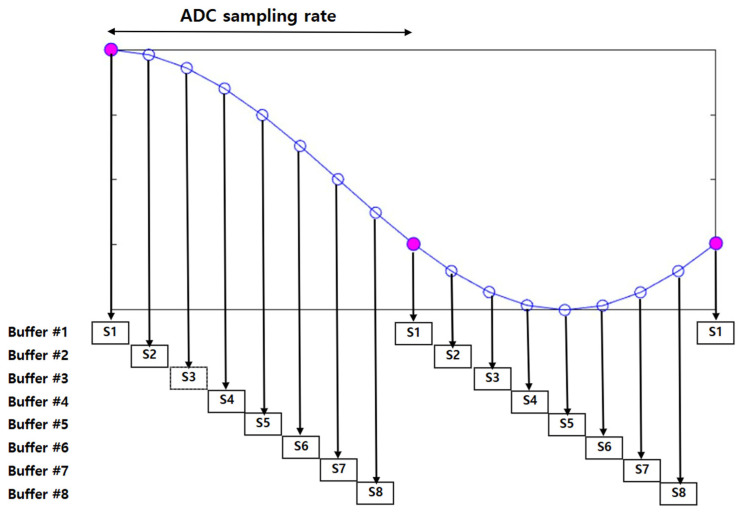
An example for sampling in offline phase.

**Figure 10 sensors-25-01844-f010:**
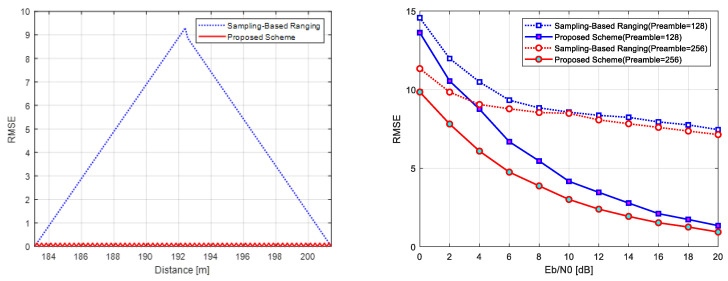
Comparison of ranging performance: ideal no-noise channel (**left**) and AWGN (**right**).

**Figure 11 sensors-25-01844-f011:**
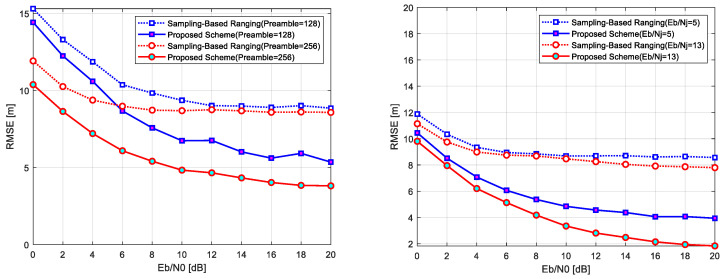
Comparison of ranging performance under jamming conditions: pulse jamming scenario (**left**) and tone jamming scenario (**right**).

**Figure 12 sensors-25-01844-f012:**
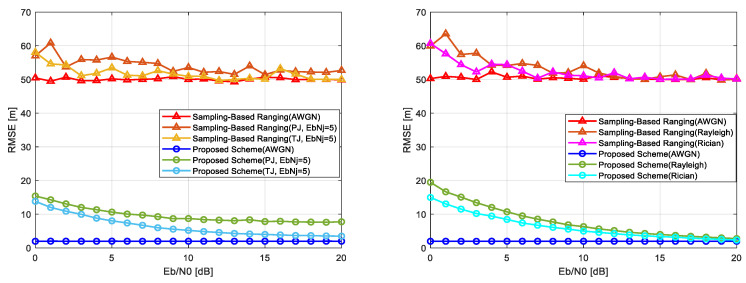
Comparison of localization performance: jamming conditions (**left**) and fading environment (**right**).

**Figure 13 sensors-25-01844-f013:**
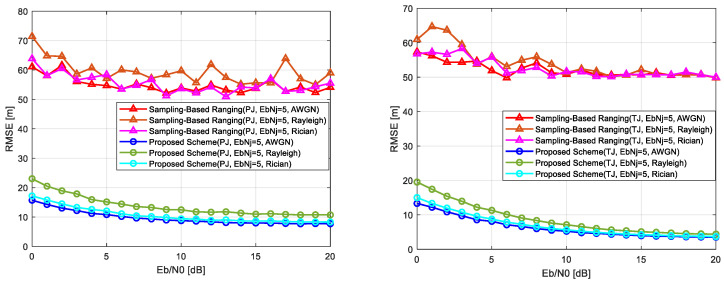
Comparison of localization error performance: combined pulse jamming and fading condition (**left**) and combined tone jamming and fading conditions (**right**).

**Table 1 sensors-25-01844-t001:** Key parameters of the proposed ranging scheme.

Parameter	Value
Chirp Duration, $T_{c}$	125 μs
Chirp Bandwidth, B	1 MHz
Preamble Sequence	Zadoff–Chu
Preamble Length	128, 256
Distance	1 km
Sampling Rate	16.384 MHz
$E_{b}/N_{j}$	5 dB, 13 dB
Pulse Duration Ratio	0.1 for 1 MHz
Channel	AWGN

**Table 2 sensors-25-01844-t002:** Key parameters of the proposed localization scheme.

Parameter	Value
$\mathrm{Chirp}\;\mathrm{Duration},\;T_{c}$	125 μs
Chirp Bandwidth, B	900 kHz
Preamble Sequence	Zadoff–Chu
Preamble Length	256
Distance	1 km
Sampling Rate	2.048 MHz
$E_{b}/N_{j}$	5 dB
Pulse Duration Ratio	0.1 for 900 kHz
Channel	AWGN, Rayleigh, Rician
Rician K-Factor	7

## Data Availability

Data are contained within the article.

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
