# Peer review of "Improvement of Wireless Localization Precision Using Chirp Signals"

_sensors, 2025, doi:10.3390/s25061844_

Round 1
Reviewer 1 Report
Comments and Suggestions for Authors
This study presents a novel localization framework that leverages the unique properties of chirp signals combined with a TDMA-based tactical data link to achieve high-precision positioning. The proposed framework integrates chirp-based ranging and localization algorithms, incorporating raised cosine interpolation and circular shift techniques to improve temporal resolution and ensure precise peak detection. This work is timely and interesting, but the following issues are suggested to address.
1. The proposed method relies on the chirp signals for wireless localization sensing. What are the advantages of this signal compared to the OFDM signals?
2. In simulations, the signal bandwidth is 1MHz, which means the range resolution is 150m. This is a large value. Why the RMSE can achieve the level of 2m?
3. How does the resolution in the time and frequency domain affect the performance of the proposed method?
4. For wireless sensing and estimation problem, the reviewer suggests to review recent works based on tensor decomposition and OFDM signals to enrich the current study, such as Channel Estimation for Movable-Antenna MIMO Systems via Tensor Decomposition, Channel-Training-Aided Target Sensing for Terahertz Integrated Sensing and Massive MIMO Communications.
Author Response
"Please see the attachment."

Reviewer 2 Report
Comments and Suggestions for Authors
This paper presents a well-structured approach to improving wireless localisation precision using Chirp 2 signals. The detailed mathematical modelling and simulation results provide theoretical backing, and the consideration of multipath interference makes this work relevant for real-world applications.
areas of improvement:
- All variables and symbols in equations should be properly defined right after they appear.
- the abbreviations TDMA and TDoA are used in the abstract without being defined first.
- The introduction includes many references ([10-19]) but it does not explain why previous methods are limited or why this approach is better. Each related work should be investigated separately.
- Can you revise lines 115-126 and provide more descriptions, if possible? The way equations (3) and (4) have been written for different message bits is a bit vague.
- How is Chirp 2 different from traditional chirp signals or CSS?
- The study only relies on Monte Carlo simulations and no hardware experiments. If hardware testing isn’t possible, you could use an existing dataset for validation.
- The paper only compares the proposed method to conventional sampling-based ranging. Could it be benchmarked against other high-precision localisation methods like ultra-wideband, hybrid, or AI-based techniques?
- A flowchart of the localisation algorithm should be added to make it easier to follow.
- A visual comparison of multipath effects with/without Chirp 2 should be added.
- Some figures especially figures 2, 3, 5, and 7 have poor quality.
Author Response
"Please see the attachment."

Round 2
Reviewer 1 Report
Comments and Suggestions for Authors
no comments
Author Response
Dear Reviewer,
Thank you for taking the time to review our manuscript. We appreciate your assessment and your confirmation that the English language does not require improvement. If you have any further feedback in the future, we would be happy to address it.
Best regards,
[Ki Tae Kim]
[Soongsil Univ.]
Reviewer 2 Report
Comments and Suggestions for Authors
The authors have addressed most of the previous comments. There are just a few minor comments left at this point:
1- thanks for your question about my earlier comments. After going back through the paper, I realised my mention of Chirp 2 was a mix-up from copying/pasting text with line numbers; apologies for the confusion! What I was aiming to ask is if you could clarify a bit more how your proposed chirp-based localization framework differs from traditional CSS approaches, especially DM-CSS and BOK-CSS, which you mention in the introduction (lines 59-76). You point out that CSS has limitations with bandwidth and sampling rates, so it’d be helpful if you could explain more clearly how your method addresses those issues, particularly when it comes to localization accuracy. To better compare your framework with traditional CSS approaches, you can add a figure/chart showing the impact of multipath interference (or jamming) on localization accuracy
2- figures 2, 3, and 7 are still a bit low in quality. Could you replace them with higher-resolution versions?
3- There are still quite a few bulk citations grouped together in the introduction section, like [12-18] . Could you split them up a little? maybe mention the key points of each, or group them by method, with a quick note on the pros/cons.
4- some references in the reference section appear to be outdated. Could you please update them with more recent studies, preferably from 2022-present, if possible?
Thank you
